# Dendrimer-Mediated Delivery of Anticancer Drugs for Colon Cancer Treatment

**DOI:** 10.3390/pharmaceutics15030801

**Published:** 2023-03-01

**Authors:** Divya Bharti Rai, Kanakaraju Medicherla, Deep Pooja, Hitesh Kulhari

**Affiliations:** 1School of Nano Sciences, Central University of Gujarat, Gandhinagar 382030, Gujarat, India; 2Department of Human Genetics, College of Science and Technology, Andhra University, Visakhapatnam 530003, Andhra Pradesh, India; 3School of Pharmacy, National Forensic Science University, Gandhinagar 382007, Gujarat, India

**Keywords:** colon cancer, dendrimers, drug delivery, passive targeting, active targeting

## Abstract

The third most common cancer worldwide is colon cancer (CC). Every year, there more cases are reported, yet there are not enough effective treatments. This emphasizes the need for new drug delivery strategies to increase the success rate and reduce side effects. Recently, a lot of trials have been done for developing natural and synthetic medicines for CC, among which the nanoparticle-based approach is the most trending. Dendrimers are one of the most utilized nanomaterials that are accessible and offer several benefits in the chemotherapy-based treatment of CC by improving the stability, solubility, and bioavailability of drugs. They are highly branched polymers, making it simple to conjugate and encapsulate medicines. Dendrimers have nanoscale features that enable the differentiation of inherent metabolic disparities between cancer cells and healthy cells, enabling the passive targeting of CC. Moreover, dendrimer surfaces can be easily functionalized to improve the specificity and enable active targeting of colon cancer. Therefore, dendrimers can be explored as smart nanocarriers for CC chemotherapy.

## 1. Introduction

Among various types of cancer worldwide, colon cancer (CC) is among the top five cancers both in males and females. It appears as a malignant tumor in the colon mucosa, which originates in the gut lumen of the intestine and, if left untreated, can spread into the muscle and serosa that lie beneath the gut wall [1]. The main risk factors include age, personal history, family history, and racial and cultural background. Only a small percentage of CC cases are linked to underlying genetic diseases; the majority are caused by lifestyle factors and aging. Overall, 10% of new cases diagnosed globally and 9.4% of cancer-related mortalities are caused by CC, estimated at around 6 lakhs fatalities each year [2]. Oncologists frequently use traditional treatment modalities such as surgery, chemotherapy, and radiotherapy to cure CC.

Surgical resection of CC involves the elimination of malignant cell mass along with a portion of adjacent normal tissue. The majority of colorectal cancer patients receive this therapy. Additionally, a portion of the healthy and any adjacent lymph nodes will be removed. Pain and discomfort in the surgical area are typically adverse consequences after surgery. Constipation or diarrhea also results after the procedure due to bowel obstruction. On the other hand, X-rays of strong intensity are employed in radiation therapy to destroy cancer cells. Fatigue, moderate skin responses, stomach distress, and loose or bloody stools are all possible side effects of this therapy. Moreover, after pelvic radiation therapy, sexual health issues and infertility (the inability to conceive) might develop. Chemotherapy is the procedure of killing cancer cells, inhibiting their growth, division, and production of new cancer cells. Typical chemotherapeutics involves systemic administration intravenously, preventing the generation from treating CC, and distributing medication to both desired (cancer site) and undesirable (normal cells) sites. Site-specific chemotherapeutics in the malignant colon zone can mitigate or eliminate these deleterious effects at unwanted sites. This would be extremely advantageous in increasing effectiveness at the site of action with no or very few negative effects. Scientists studying medication delivery are looking into ways to make drugs more permeable through the colonic epithelium, which might help to administer chemotherapy drugs both locally and systemically [3].

Nanoparticles (NPs) are being clinically investigated as smart nanocarriers for localized CC chemotherapy, additionally lowering the doses and increasing the drug’s aqueous solubility and bioavailability with fewer side effects [4,5,6]. However, biodegradable nanoparticles such as micelles, nanogels, liposomes, nanoemulsions, and polymeric NPs have issues of burst release, aggregation, and less stability [7]. On the other side, inorganic NPs such as metal NPs, carbon-based NPs, and silica NPs have a low drug-loading capacity, slow clearance, and less scope for functionalization [8]. In comparison to these drug carriers, dendrimers can be used as carriers due to their unique physicochemical features and biodegradable backbones [9]. Dendrimers are monomeric, uniformly hyperbranched nanomaterials with arms or branches that resemble trees and originate from a central core [10,11]. There are several anionic, neutral, or cationic end groups on the arms that can be further decorated with multiple functional groups for a particular application on the dendrimer’s outer surface [12,13,14]. A dendrimer generation is referred to monomeric branches that are added to the core during the synthesis [15]. With increasing generation, dendritic macromolecules typically expand in size and take on a more globular form, thereby offering large void spaces for encapsulation [16]. Hydrogen bonds, chemical linkages, or hydrophobic interactions can be involved in loading drugs into dendrimer core cavities and conjugate targeting ligands onto the surface endings [10]. By creating functionalized dendrimers for targeted therapy, the chemotherapeutic efficacy of CC drug improves, and its systemic toxicity reduces [17]. This review focuses on using dendrimers as a smart nanocargo for the anticancer drug delivery against CC.

## 2. Anticancer Drugs for the Treatment of Colon Cancer

Generally, anticancer drugs interfere with crucial cellular processes, including adhesion, invasion, migration, proliferation, cell death, cell division, angiogenesis, self-renewal, and drug resistance. On the molecular level, these drugs work by either activating/increasing or inactivating/decreasing various metabolic molecules such as cell-signaling factors, genes, and proteins [18]. These therapeutic agents can be broadly classified as synthetic and natural drugs. More than 33 formulations are permitted by the Food and Drug Administration (FDA) for CC chemotherapy. The broader medication of CC includes synthetic drugs such as 5-fluorouracil (5-FU), capecitabine (CPB), irinotecan (IR), and oxaliplatin (OX), which show strong antitumor efficacy in CC (Figure 1). These drugs have also been used in combinations, including the FOLFOX regimen, FOLFIRI regimen, and XELOX, with or without a monoclonal antibody agent, for enhanced anticancer activity [19]. Moreover, several natural compounds, including curcumin (CUR), piperlongumine (PPL), resveratrol (RES), quercetin (QUE), and gallic acid (GA) have clinically being studied as successful therapeutic molecules against CC (Figure 1) [20].

### 2.1. Natural Drugs

In order to treat CC more effectively and with few adverse effects, plant-isolated products can be helpful as the existing medicines typically have negative effects. A literature search led to the inclusion of 172 research items and 71 trials involving 190 herbal species. The major species for preventing CC include grapes, soybean, green tea, garlic, olive, and pomegranate. Extract from every part including fruits, seeds, leaves, and roots was investigated with in vitro and in vivo models [21]. Global statistics from the World Health Organization (WHO) implies that 75–80% of the individuals exclusively use conventional medical systems as their first line of therapy because of the safety of natural drugs. Additionally, references to the use of plants for treating various diseases in religious texts drew additional scholars who focused on assessing the scientific veracity of traditional claims [20].

#### 2.1.1. Curcumin

Curcumin (CUR), a polyphenolic compound chemically named diferuloylmethane, is a native ingredient of an ancient Indian spice known as *Curcurma longa.* Three main chemopreventive mechanisms of CUR against various types of cancer, including CC, have been reported, i.e., antioxidant mechanism, apoptosis, and cell cycle block. CUR is a highly lipid-soluble oxygen-scavenging compound that diffuses into cell membranes and reaches the endoplasmic reticulum, mitochondria, and nucleus. The hydroxyl and methoxy groups of CUR then directly chelate to the superoxide anions, H_2_O_2_, nitrite, and reactive oxygen species (ROS) and reduce the oxidative stress [22].

Additionally, CUR induces apoptosis by inhibiting cyclooxygenase-2 (COX-2), Phosphorylated Ak strain transforming (pAKT) kinase, and increases the p-AMP protein kinase (pAMPK) signal. It has been found CUR can upregulate the death receptor 5 (DR5) protein, a receptor that is essential for apoptosis in CC cells, such as HCT-116 and HT-29. CUR also activates caspase 8, which starts the Fas-mediated apoptotic pathway. The death-inducing signaling complex (DISC), created by procaspase 8 and the Fas ligand, activates caspase 8 through the degradation and activation of caspase 3, caspase 7, and *BH3 Interacting Domain Death Agonist* (Bid) [23].

The third mechanism slows the spread of malignancies by blocking signaling pathways, including STAT3, epidermal growth factor receptor (EGFR), and nuclear factor kappa B (NF-κB) that support tumor growth and metastasis (Figure 2). Hence, patients with colon pre-cancerous polyps may benefit from CUR. It prevents NF-κB, the main transcription factor involved in tumor progression, from being activated by tumor necrosis factor (TNF) [24]. It has also been observed that CUR prevents the binding of the phosphoryl group and activation of the p65 component of NF-κB, thereby stopping its transport in the nucleus by initially blocking phosphorylation and degradation by an inhibitor of nuclear factor kappa B (IF-κB). This is probably caused by methoxy groups located on the ortho position of the phenyl ring in the CUR molecule. By inhibiting the NF-κB pathway, CUR alters several tumor-related genetic materials, such as genes and miRNAs. This, in turn, inhibits the production of pro-metastatic enzymes such as matrix metalloproteinases (MMPs) and COX-2 and pro-inflammatory cytokines [25]. These anticancer mechanisms of CUR result in the reduction of tumor size and number; and prevent metastasis and tumorigenic mutations.

#### 2.1.2. Piperlongumine

Piperlongumine (PPL), also known as piplartine (5,6-dihydro-1-[(2E)-1-oxo-3-(3,4,5-trimethoxyphenyl)-2-propenyl]-2(1H)-pyridinone), is a naturally occurring amide alkaloid discovered in long pepper, *Piper longum* L. The compound has been reported to have chemopreventive action in the case of lung, breast, liver, kidney, pancreatic, stomach, colon, bladder, lymphoma, melanoma, oral, head, and neck cancer [26]. There are three ways of working PPL; by raising ROS levels and decreasing antioxidant enzyme activity, activating the Suppressor of Mothers against Decapentaplegic family member 4 (SMAD4) pathway, and by antimetastatic action.

In malignant cells, PPL specifically increases the generation of ROS. An excessive amount of oxidative stress triggers apoptosis, which leads to cell death. PPL generates ROS, which successively blocks Akt activation and induces cancer cell death through the caspase- or Ras-mediated pathway. In another mode, SMAD4 overexpression increases the production of p21 (Senescence-inducing p21 is a crucial cell cycle regulator), which blocks regulating B-cell lymphoma 2 (BCL2) and the anti-apoptotic protein survivin and causes apoptosis. Hence, the ROS/Ras/Akt and SMAD4 signaling pathways are implicated in the tumor-killing activity of PPL in the colon (Figure 3) [27].

Sometimes PPL stimulates p38 mitogen-activated protein kinases, Akt, Jun N-terminal kinase (JNK), and extracellular signal-regulated kinase (ERK), which encourage protein glutathione or NF-κB inhibition. Akt deactivation decreases cyclin D level and ceases the cell cycle at the G2/M phase, thus stopping tumor enlargement [28]. Additionally, the down-regulation of the genes fibronectin-1 (FN1), cadherin-2 (CDH2), catenin beta-1 (CTNNB1), and twist-related protein 1 TWIST1 is a well-known mechanism of PPL to control cancer cell metastasis [29].

#### 2.1.3. Resveratrol

Resveratrol (RES) (trans-3,4′,5,-trihydroxystilbene) is a stilbenoid that comes under the polyphenolic class of compounds present in numerous plants, including peanuts, grapes (*Vitis vinifera*), mulberries (*Arachis hypogaea*), tea and Itadorii plant. It consists of three hydroxyl groups, two aromatic rings, and an ethylene bridge. RES can exist in cis and trans isomeric forms, where the trans form generally predominates in nature. Exposure to UV light, pH above 6.8, and heat over 37 °C converts the trans form to less stable cis, which quickly degrades [30]. The anticancer property of RES has mostly been credited to its antiproliferative, antimetastatic, and apoptotic induction [31].

According to reports, RES promotes programmed cell death in CC cells by activating the serine-threonine kinase (AKT)/STAT signaling system. The PI3K/AKT pathway is crucial to stop cell death and damage. AKT deactivation can initiate apoptosis by inhibiting caspase-9, caspase-8, and subsequently caspase-3 phosphorylation and BCL2-associated agonist of cell death (BAD). RES binds through hydrogen bonding and inactivates the ATP-binding site of AKT1 and AKT2. Other than this, the STAT3 (signal transducer and activator of transcription 3) signaling pathway is a crucial factor in DNA synthesis that RES uses to reduce the growth of colon cancer cells. Therefore, blocking this signaling pathway may be important for the treatment of CC. The downregulation of AKT suppresses several proteins such as surviving, cellular FLICE (FADD-like IL-1β-converting enzyme)-inhibitory protein) inhibitory protein (cflip), and cellular inhibitor of apoptosis proteins (cIAPs) while elevating BCL2-associated X (Bax) in a STAT-3 independent manner [32]. AKT1/2 knockdown by RES also ceases cell cycle at the beginning of S-phase by lowering the level of cyclin-D1 and cyclin-E2 due to activation of the p53 tumor suppressor gene.

The levels of ROS can be controlled by RES due to its capacity to suppress its production, lipid peroxidation, and the modification of antioxidant-associated enzymes. Superoxide dismutase (SOD), catalase (CAT), glutathione reductase, glutathione peroxidase (GPX), and glutathione S-transferase are some of the enzymes involved in the antioxidant mechanism. Non-enzymatic mechanisms are mediated by reduced glutathione, ascorbic acid, α-tocopherol, and vitamin A. Manganese superoxide dismutase is a crucial antioxidant agent that breaks down ROS into molecular oxygen and hydrogen peroxide. By reducing the production of this enzyme, RES aids in the suppression of harmful free radicals. By upregulating SOD, CAT, GPX, and activating the sirtuin (silent mating type information regulation 2 homolog) 1 Sirt1/AMPK and nuclear factor erythroid 2–related factor 2 (Nrf2) pathways, resveratrol has already demonstrated suppressive effects on peroxynitrite-mediated proteins and lipids oxidation [33]. Inhibition of ERK1/2 and activation of MAPK are other mechanisms by which RES regulates ROS levels to block the p38 Mitogen-activated protein kinase (MAPK) signaling pathway that induces COX-2 and iNOS (isozyme of nitric oxide synthase) [31,34].

#### 2.1.4. Quercetin

Quercetin (QCT) (3,3′,4′,5,7-pentahydroxyflavone) is a lipophilic phenol plentiful in fruits, vegetables, seeds, berries, and tea. QCT has the capacity to function as a chemopreventer and can suppress the proliferation of cancer cells. Its ability to prevent cancer has been linked to signaling events involving antioxidant activity, cell-cycle arrest, and cell death. Apoptotic, p53, NF-κB, MAPK, janus kinase (JAK)/STAT, phosphatidylinositol-3 kinase (PI3K)/AKT, and wingless-related integration site (Wnt)/catenin pathways are among the molecular process regulated by QUE. It upregulates two cell cycle regulatory proteins, p21WAF1, and p27KIP1. Additionally, quercetin inhibits the activity of tyrosine and serine-threonine kinases, which are connected to pathways for survival, including MAPK and AKT/protein kinase B (PKB). QUE downregulates the activity of oncogenes (H-ras, C-myc, and K-ras) while activating tumor suppressor non-coding RNAs (ncRNAs). Quercetin also influences the expression of different miRNAs [35].

An important signaling pathway NF-κB is abnormally activated in many different forms of cancer and regulates the transcriptional activation of genes crucial for the strict regulation of several cellular activities. NF-κB pathway inhibitors have demonstrated potential antitumor properties. Tumor necrosis factors (TNF) or other TNF superfamily members can cause cancer cells to die, but NF-B, a crucial apoptosis inhibitor, can prevent this from happening. QUE causes apoptosis in human CC by blocking the NF-κB pathway and upregulating pro-survival factors such as Bax and down-Bcl-2 as a result of IF-κB degradation. Caspase signaling starts when the Bax/Bcl-2 ratio changes. In other words, QUE raised the Bax/Bcl-2 ratio, which in turn caused caspase-3 to become activated. QUE administration results in proteolytic cleavage of poly (ADP-ribose) polymerase (PARP) reactant for caspase-3, which triggers an apoptotic signaling cascade [36].

QUE’s antitumor action through the Wnt/catenin pathway has also been shown in several studies. QUE reduces survivin, which is important in cell cycle control and death. Phosphatidylinositol-3-kinase (PI3K) is required for AKT to translocate to the plasma membrane of the cell. The PI3K/AKT pathway, which has anti-apoptotic capabilities (a pro-apoptotic gene), regulates the Bcl-2 protein family and Bax. This pathway’s deregulation may be a crucial step in the development of cancer. Numerous research works have investigated the impact of quercetin on the PI3K/AKT pathway. QUE may lower active AKT/PKB phosphorylation and substantially inhibit cell proliferation [37].

The Janus kinase/signal transducers and activators of transcription (JAK/STAT) signalling pathway receives stimuli signals from the outside of the cell and delivers its message to the cell nucleus, activating transcription factors. The JAK/STAT pathway controls the maintenance of the immune system, cell growth and division, cell death, and tumor development. Other pathways, including ERK, MAPK, and PI3K, control JAK/STAT pathway components. QUE also produces intracellular ROS, lowers mitochondrial membrane potential, and stimulates sestrin 2 expression via the AMPK/p38 pathway [37].

QUE has been demonstrated to activate the p53-ROS pathway, lower the production of cyclin-D1, and cause epigenetic alterations that stop the cell cycle in the S phase. Apoptosis-related proteins such as AMPK, p53, and p21 were upregulated by QUE, which also blocks mitosis in the G1 phase. In different kinds of cancer cells, AMPK activation controls apoptosis, and p53 upregulation is linked to a considerable elevation of AMPK phosphorylation. QUE led to the phosphorylation of AMPK and the induction of phospho-p53 [38].

#### 2.1.5. Gallic Acid

Gallic acid (GA), chemically known as 3,4,5-trihydroxybenzoic acid (C_6_H_2_(OH)_3_COOH), is a dietary phenol being studied due to its anticancer properties. It is extracted from a wide number of plants, such as oak, bearberry, blackberry, Indian gooseberry, raspberry, vinegar, wine, and green tea [39].

GA causes ROS-dependent apoptosis and prevents colon cancer cells from proliferating. GA also causes cell cycle arrest and early apoptotic processes such as lipid layer rupture and a decrease in mitochondrial membrane potential [39]. Additionally, by increasing the ratio of degraded caspase-3/pro-caspase-3 and degraded caspase-9/pro-caspase-9, GA reduces cell proliferation and increases cell apoptosis. GA simultaneously decreases the levels of (p)- proto-oncogene tyrosine-protein kinase (SRC), (p)-EGFR, (p)-AKT, and (p)-STAT3 [40].

### 2.2. Synthetic Drugs

To date, 16 approved formulations are available in the market for the treatment of CC. Out of these, 12 molecules are used as primary chemotherapeutic agents. These include 5-fluorouracil (5-FU), capecitabine (CPT), irinotecan (IR), oxaliplatin (OX), leucovorin, regorafenil, trifluridine, tipiracil, cetuximab, aflibercept, panitumumab, and bevacizumab [41].

#### 2.2.1. 5-Fluorouracil

5-FU a cyclic compound, is an analog of uracil with a fluorine instead of hydrogen at the C-5 position is similar to uridine nucleotide base of DNA and RNA [42]. Therefore, it can be incorporated into RNA and DNA, interfering with nucleic acid synthesis. Hence, DNA replication is suppressed by the synthetic fluorinated pyrimidine analog 5-FU and causes cytotoxicity [43]. There are many levels at which 5-FU may inhibit CC.

Firstly, 5-FU inhibits thymidylate synthase (TS), the enzyme which catalyzes thymidylate synthesis [44]. To sustain DNA replication and repair, deoxyuridine monophosphate (dUMP) is methylated by TS to deoxythymidine monophosphate (dTMP), with 5,10-methylenetetrahydrofolate (CH_2_THF) acting as the methyl donor. 5-FU instead gets converted into fluorodeoxyuridine monophosphate (FdUMP), which inhibits the synthesis of dTMP by forming a stable complex with TS. Declining dTMP production impair cell division since it is required for DNA synthesis and repair. The rate-limiting phase of 5-FU production in both normal and malignant cells is the transformation of 5-FU to dihydrofluorouracil (DHFU), which is mediated by the enzyme dihydropyrimidine dehydrogenase (DPD). The ternary TS-FdUMP-CH_2_THF complex forms and accumulates the fluorine is unable to separate from the pyrimidine ring, progressively inactivating the enzyme. Deoxythymidine triphosphate (dTTP) is depleted as a result of the depletion of dTMP, which imbalance the ratio of the other deoxynucleotides, deoxyadenosine triphosphate (dATP), deoxyguanosine triphosphate (dGTP), and deoxycytidine triphosphate (dCTP). Finally, it is believed that change in the ATP/dTTP ratio seriously impairs replication and repair, leading to fatal DNA mutation. Misintegration of dUMP, results in higher amounts of deoxyuridine triphosphate (dUTP), and 5-fluorouridine triphosphate (FdUTP), the byproduct of 5-FU also effect DNA replication. Additionally, it has been hypothesized that the repairing enzyme uracil-DNA-glycosylase (UDG) is inactive due to elevated FdUTP/FdTTP ratios and promotes additional erroneous DNA repair [45].

Secondly, since 5-FU is analogous to uracil, it may also be mistakenly integrated into RNA, and research indicates that its cytotoxicity is significantly influenced by RNA-based processes. Research suggests that abnormalities in the ribosomal RNA processing component (rrp6) may contribute to 5-FU hypersensitivity [46].

According to other findings, ribosomal RNA (rRNA) maturation is a crucial target for 5-FU. Additionally, exosome-dependent accumulation of polyadenylated rRNAs has been demonstrated to be enhanced by 5-FU. Additionally, there is proof that 5-FU interferes with premature RNA splicing by affecting U2 spliceosomal RNA (U2 snRNA) pseudouridylation. Furthermore, 5-FU blocks the formation of snRNA/protein complexes and post-transcriptional alteration of transfer RNA (tRNA), which prevents pre-mRNA splicing. The most prevalent post-transcriptional alteration of non-coding RNA pseudouridylation can also be inhibited by RNA that contains the 5-FU molecule. According to the data, putative pseudouridine synthase (Cbf5p) binds firmly to substrates containing 5-FU and causes the Trf4/Air2/Mtr4p polyadenylation (TRAMP)/exosome-mediated RNA surveillance pathway to degrade those substrates. Additionally, Cbf5p pseudouridylation activity is required for RNA-based 5-FU toxicity, and Cbf5p-dependent 5-FU toxicity was decreased when it was sequestered to a specific guide RNA [47].

#### 2.2.2. Capecitabine

Despite the activity of 5-FU CC, CPB was formulated as its prodrug due to its short half-life and demand for continual infusions in the case of 5-FU. Moreover, it can be given orally and hence is more patient-compliant. Hence, the FDA authorized CPB under the trade name Xeloda^®^ as an oral prodrug of 5-FU in June 2005 [48].

CPB is a fluoropyrimidine carbamate with antineoplastic action used to treat CC that has spread to other body parts. It is a systemic prodrug that has limited pharmacologic efficacy, however changed into 5-FU. 5-FU subsequently converted to FdUMP and FUTP by TP, which were then taken up by both normal and malignant cells. TP has expressed more in many tumors than in normal tissues or plasma and preferentially activates the prodrug CPB to 5-FU at tumor site. Therefore, patients who express more TP and are resistant to 5-FU will respond to CPB more favorably. Within both normal and tumor cells, 5-fu is further converted to two active metabolites, FdUMP and FUTP. These metabolites derived from CPB harm CC cells in two ways, similar to 5-FU. First, the TS-FdUMP-CH_2_THF complex form which prevents the conversion of dUMP to dTTP. Lack of dTTP might inhibit cell division. Second, nuclear transcriptional enzymes may unintentionally switch FUTP instead of UTP during the synthesis of RNA. Moreover, this metabolic error might impede the processing of RNA and protein production by producing incorrect RNA [49,50].

#### 2.2.3. Irinotecan

IR is a CPB analog that prevents topoisomerase I (topo I) from working. In October 2015, IR was authorized for the treatment of advanced pancreatic cancer as liposome injection under the trade name Onivyde [51,52]

IR and its active metabolite SN-38 (7-ethyl-10-hydroxy-camptothecin (CPA)) adhere to the topo I complex itself forming a ternary complex with topo I and DNA. This complex blocks the replication fork and renders nick repair in the DNA strand thereby encouraging cell apoptosis and damage of DNA duplex. Actually, topo I causes reversible single-strand breaks throughout these processes, enabling single DNA strands to run along the break and relieving torsional tension in the DNA. The topo I form a covalent bond with the 3′-DNA terminal of the broken DNA strands to create a catalytic intermediate known as a cleavable complex. Topo I rejoins the cleaved DNA duplex to create the relaxed DNA conformation required for DNA synthesis and the strand passage reaction has finished. These single-strand breaks cannot be repaired because of the binding of IR to the topo I-DNA complex and unrepaired DNA eventually leads to cell apoptosis NotablyAlso, the IR-associated anticancer mechanism is particular to the S-phase of the cell cycle. Overall, IR decreased levels of topo I expression, changes conformation of topo I due to various mutations, changes to the response to IR-topo I-DNA complex, including topo I being degraded by the proteasome and leading to DNA repair, and activation of the NF-κB [53]. IR is also given for combinational therapy along with 5-FU as a formulation named FOLFIRI (leucovorin, 5-FU, and IR).

#### 2.2.4. Oxaliplatin

CC can be treated with the platinum-based chemotherapy medication oxaliplatin (OX). It is most frequently coupled with leucovorin, a folinic acid, and 5-FU. The primary therapy regimen for CC has used a combination of these medications, known as fluorouracil/leucovorin calcium/oxaliplatin folinic acid (FOLFOX) and CPT/OX (XELOX). Because it has a 1,2-diaminocyclohexane (DACH) ligand, oxaliplatin’s molecular structure sets it apart from other platinum-based chemotherapy medicines. Together with its platinum ingredient, DACH makes DNA more difficult to repair, increasing its capacity to destroy tumor cells. The nucleotide excision repair (NER) pathway is connected to oxaliplatin resistance. OX resistance is connected with the expression levels of genes involved in excision repair, which are predictive indicator for drug sensitivity. Numerous cells inside the total mesorectal excision (TME) release large amounts of transforming growth factor-1 (TGF-1). Through the epithelial-to-mesenchymal transition (EMT), TGF-1 is hypothesized to aid in the induction of resistance to oxaliplatin. Thus, blocking TGF-1 to prevent EMT may make tumor cells more susceptible to OX-mediated cell death [54].

## 3. Role of Dendrimers in the Delivery of Anticancer Drugs against Colon Cancer

A distinct family of polymeric macromolecules called dendrimers was discovered by Volte and Tomalia et al. between 1970 and 1990 [55]. Dendrimers are created by a series of synthesis steps where the repetition of monomeric units is incorporated to produce dendrimers with several generations. A generation is a level of additional branches introduced to the surface of dendrimers throughout the synthesis process. Dendrimer synthesis can be accomplished by divergent or convergent growth strategies. The synthesis in the convergent growth process starts with the individual branches, which are then joined to form a single dendrimer structure. They are three-dimensional hyperbranched structures ranging from 1 to 100 nm in size. The architecture comprises a central core, with arms or branches that resemble trees originating from the core with at least one branch junction and terminal functional groups at the periphery [56]. These end groups on the dendritic arms could be anionic, neutral, or cationic. With increased dendrimer branching, dendritic macromolecules typically linearly expand in size and take on a more globular form. Contrary to linear polymers, dendrimers have unique polymeric properties such as regulated size, molecular weight, shape, and monodispersity [57]. Moreover, dendrimers are great nanocarriers with interior molecular cavities and peripheral functional groups to encapsulate or conjugate molecules [58]. Hydrogen bonds, chemical linkages, or hydrophobic interactions can all be used to load the active pharmaceutical ingredients and attach targeting ligands on the surface of dendrimers [59]. The interior core and end terminals can be modified according to the applications. Moreover, certain pharmacochemical properties of dendrimers, including solubility, stability, biodegradable backbones, and penetration ability, make them the ideal candidate for the delivery of biomolecules such as drugs, genes, or bioimaging agents [60,61]. Dendrimers generate organic or inorganic hybrid nanoparticles that self-assemble and stabilize them [9]. Additionally, the numerous positively-charged groups on the surface of dendrimers interact with the negatively charged cellular membrane and contribute to cytotoxicity, especially for cancer cells [62].

Dendrimers are available in a huge variety and may be categorized based on their structure, branching, solubility, terminal moieties, and generation [63]. Polyamidoamine (PAMAM), poly(propyleneimine) (PPI), and poly(L-lysine) (PLL) are the most explored dendrimers for anticancer drug delivery [47,49]. Due to its simple synthesis, and excellent physicochemical characteristics, PAMAM outperforms other dendrimers for anticancer drug delivery applications [64]. Other than the PAMAM dendrimers, PPI dendrimers are examined the most for therapeutic purposes, particularly in the delivery of anticancer drugs. The entire generation is composed of primary amines, and the surface end contains a variety of tertiary tris-propylene amines [60]. PLL dendrimers are another prominent form of dendrimer that has been utilized as the delivery system for anticancer drugs [65]. Similar to PAMAM dendrimers, they are flexible and highly aqueous soluble, biocompatible, and biodegradable. The use of dendrimers in anticancer drug delivery further depends on functionalization. Multiple active sites are added to dendrimers by the process of functionalization, which results in macromolecules with multifunctional architecture [66]. Six different characteristics, including size, structure, surface chemistry, flexibility, stiffness, architecture, and elemental composition, referred to as “Critical Nanoscale Design Parameters” (CNDPs), are considered while functionalizing dendrimers [67]. These characteristics may be altered following the therapeutic use of the dendrimer nanoparticles, altering the inherent characteristics, function, and performance of the nanoparticles [68]. There are generally three purposes of dendrimer surface modification. The first priority is their high biocompatibility and limited toxicity [66]. The second focus is on enhancing surface chemistry to introduce multiple pharmacological advancements in dendrimers and to specifically target certain cells [60]. The third purpose of surface functionalization focused on developing stimuli-responsive, or “smart” nanoparticles that can respond to biological and environmental stimuli (such as temperature, light, pH, etc.). Dendrimer nanoparticles can be functionalized to increase biocompatibility (e.g., by PEGylation or acetylation), increase permeability (typically, by functionalizing amino acids or lipids), or induce site-specific delivery (e.g., with aptamers, antibodies, vitamins, or peptides) while designing it as an anticancer drug carrier [69,70]. Overall, dendrimers offer the following advantages in anticancer drug delivery to improve their therapeutic performance (Figure 4):i.Solubility increment—Being highly water soluble itself, dendrimers solubilize anticancer drugs, which are commonly hydrophobic, hence, improving the bioavailability of the drug [71]. Morgan et al. improved the solubility of poorly water-soluble CPA (~20 μmol/L) by encapsulation in Poly (glycerol succinic acid) dendrimers. The amount of CPA was determined to be 240 μmol/L in the dendrimer formulation, which was ~10 times higher than the solubility of CPA in water [72].ii.Stability—By encapsulating less stable, heat-liable, pH-sensitive, or photosensitive anticancer compounds, dendrimers prevent degradation and increase the storage stability of such drugs. Moreover, surface-charged dendrimers may prevent drug molecules from being aggregated through steric repulsion. In a trial, The bow-tie dendrimer’s branches break slowly due to the steric impedance of ester bonds and as a result, took many months to breakdown entirely. The bow-tie dendrimers’ slow breakdown of bow tie dendrimer improves the stability of the encapsulated drugs in the systemic circulation [73].iii.Permeability—Dendrimers may be used to enhance membrane permeability and, therefore, cellular uptake of anticancer drugs [74]. Teow et al. designed a dendrimer nanosystem to bypass the biological barriers, thereby increasing the permeability of paclitaxel (PTX) and overcoming cellular barriers. Lauryl chains were conjugated on the G3-PAMAM surface by cross-linking with glutaric anhydride. In comparison to unencapsulated PTX, transepithelial electrical resistance assay of lauryl PAMAM conjugate showed a 12-times enhancement in the transport of conjugated PTX from basolateral to apical and apical to basolateral side of the cell [75].iv.Biocompatibility—Dendrimers are biopolymers and possess biodegradable backbones, and are considered biocompatible and safer for in vivo applications. Moreover, the cytotoxicity of amine-terminated dendrimers can be rendered by surface functionalization. For example, amine terminals of dendrimers were functionalized using PEG based on a polyester-polyamide hybrid for the delivery of DOX for the treatment of CC [76].v.Prolonged circulation—Due to the tiny size, dendrimers have delayed clearance through the reticuloendothelial system (RS); hence, the encapsulated anticancer drug exceeds the circulation half-life [77]. However, due to high cationic surface charge of certain dendrimers, they are eliminated from the body. The positively charged terminals must be neutralized to increase the circulation time. The dendrimer terminals can be altered using a variety of techniques, including PEGylation, vitamin conjugation, glycosylation, triazine and hydrazone linking, acetylation, and amino acid or peptide addition and nucleic acid complexation. PEGylation is the easiest and most efficient technique to modify the dendrimer’s surface [78]. As mentioned earlier, pegylated “bow-tie” polyester dendrimers synthesized possessing cleavable carbamate bonds also increased the circulation time of dendrimers [75]. Similar to PEGylation, dendrimers treated with hyaluronic acid (HA) polymer have consistently shown the potential to lengthen systemic retention periods with altered tissue distribution. Qi et al. grafted HA on PAMAM dendrimers for extended systemic circulation of encapsulated topotecan hydrochloride. HA being hydrophilic, forms a hydrophilic coating on the PAMAM surface that might hide the surface charge and prevent PAMAM molecules from opsonization, thereby prolonging the circulation time of TPT-loaded PAMAM [79].vi.Controlled release—Dendrimers usually range from 1 to 100 nm in size and thus provide controlled drug release and optimum pharmacokinetics [80]. Gillies et al. conjugated doxorubicin (DOX) to polyester G4 dendrimers via hydrazone (hyd) cross-link which hydrolyzes at acidic pH. Dendrimer-hyd-DOX conjugates were stable at normal physiological pH (pH 7.4) and only released 10 percent of encapsulated DOX from the dendrimer system. In contrast, the release was observed to be elevated up to 100% after 48 h incubation at a pH similar to the tumor microenvironment (pH 5.0). The hydrazone-linked dendrimers enabled the pH-mediated controlled release of DOX [81].vii.Cancer targeting—Dendrimers have terminal groups that can be functionalized with an ample range of molecules according to the specificity of the targeted cancer site for a particular anticancer drug [82]. Location-specific medication delivery to the colon boosts the amount of the medicine at the target site, requiring a lower dosage and reducing adverse effects. When coupled with certain antibodies, dendrimers are more sensitive and effectively identify circulating tumor cells. For example, sialyl Lewis X antibody-conjugated PAMAM dendrimers have recently been used for targeting to precisely bind and capture HT-29 CC cells [83]. In another example, telodendrimer modified with cholic acid and vitamin E were developed for targeted delivery of gambogic acid (GBA). The GBA-telodendrimer formulation was injected into animal models, and in vivo imaging was done. The fluorescent signal indicated that the modified telodendrimers could efficiently target the xenograft model of HT-29 colon cancers. The trial on animal models administered with plain telodendrimers showed more nonspecific uptake and on contrary less uptake at the cancer site [84].viii.Multipurpose dendrimers—Numerous end groups on the surface of a dendrimer molecule provide multiple conjugation sites for different functionalization moieties to introduce multiple advancements in a single dendrimer-based nanocarrier [85] [86]. Such dendrimers serve more than one application, such as drug delivery, gene delivery, bioimaging, and cancer targeting simultaneously [87,88]. Pishavar et al. have used PAMAM for simultaneous drug delivery and gene therapy. CC cells were co-delivered in vitro and in vivo with DOX and plasmid expressing TRAIL, using G5 PAMAM functionalized with cholesteryl chloroformate and alkyl-PEG. The results demonstrated that customized PAMAM complexed with TRAIL plasmid and loaded with DOX had a higher anticancer impact than modified PAMAM carrying DOX and TRAIL plasmid separately [89]. In another experiment, the dendrimer was hybridized with gold nanorod for combined cancer photothermal chemotherapy. PEGylated-G4 PAMAM was covalently linked to mercaptohexadecanoic acid-functionalized gold nanorod and loaded with DOX. The combined treatment of cancer cells using a dendrimer-gold hybrid demonstrated higher efficacy than a single therapy module [90].

## 4. Dendrimer-Based Passive Targeting to Colon Cancer Cells

Dendrimers are used in passive targeting to extravasate and accumulate in the tumor tissue rather than healthy tissues. When it was discovered that NPs have the propensity to aggregate in cancerous cluster, the application of nanoparticles as a medication system started in 1986. The “Enhanced Permeability and Retention” (EPR) effect, commonly termed as passive accumulation or primary targeting, is a key concept in the use of dendrimers to treat cancer, more specifically, solid tumors. Because of their high selectivity, accumulation in tumor areas, and lengthy blood circulation times, dendrimers have great promise for application as cancer therapeutics [91].

There are physiological and anatomical factors underlying the hypothesis of increased permeability and retention in tumors. Angiogenesis is triggered when tumor cells multiply, group together, and form a clump of 2–3 mm. In order to meet the rising need for nutrients and oxygen in the tumor, new blood vessels are formed. These vessels have very different anatomy from that in the normal tissues. Firstly, to cater to the excessive nutrition supply in rapidly growing cancer cells, the blood vessels are enlarged and dilated with broadened lumen causing a rapid flow of blood. Secondly, the endothelial cells of the neovasculature are distorted in shape, loosely organized, and irregularly aligned with unusual pores, resulting in leaky vasculature and outflow at the tumor location. Thirdly, these blood vessels lack a basement membrane which is composed of a smooth muscle layer and lines the vessel, hence having poor lymphatic drainage [92].

Nanosized anticancer payloads such as dendrimers exploit the leaky vasculature and slow lymphatic removal in the tumor location. Any circulating NPs, such as drug-loaded dendrimers, could leak along with blood into the tumor tissue easily. Furthermore, dendrimer macromolecules reside in the tumor for a long time during extravasation into the tumor interstitium due to the narrow venous outflow in tumor tissue and poor lymphatic clearance. Within 1–2 days, very high local concentrations of drug payload, such as 10–50 times those seen in normal tissue, can be delivered at the tumor site using dendrimer macromolecules. As a result of their quick diffusion into the bloodstream and subsequent renal clearance, it is interesting to note that low-molecular-weight medicines are exempt from the EPR effect. Macromolecules with a molecular weight greater than the renal filtration range, i.e., 40 kDa, tend to preferentially aggregate in malignant tissues. If nanoparticles can bypass the RS and defy renal clearance due to their macromolecular size staying in circulation for at least six hours, they can reach threshold concentrations within the neoplastic tissue via the EPR effect [93].

Benefits of dendrimer-mediated delivery of CC drugs include improved permeability and retention of drug in the tumor, extended half-life, and lower antigenicity. Drugs become stable in the cancer site for a long duration due to NP accumulation caused by the EPR effect (Figure 5). Dendrimers being macromolecules with a high molecular weight surpasses encapsulated low-molecular-weight medicines from RES removal. Additionally, sugar- or lipid-coated dendrimers may suppress the antigenicity in the case of immunostimulatory anticancer drugs and reduces their elimination by opsonization through RES or macrophages. As a result, the half-lives of drugs loaded in dendrimers in the bloodstream can be significantly prolonged [94].

One of the most used and effective anticancer medications in CC is CPB. But CPB has some adverse effects, such as damage to the liver, bone marrow, blood, and hair cells. In this regard, Nabavizadeh et al. used G4 PAMAM dendrimers to enhance the anticancer activity of CPB and lessen its non-specific effects on organs other than the colon such as liver and bone marrow. The researchers examined the impact of the unconjugated and dendrimer-delivered CPB on tumor reduction in mouse and aberrant blood cell lines without conjugating any targeting ligand. Compared to the free version, the results revealed a more prominent reduction in the tumor size and fewer adverse effects in the liver and blood [95].

## 5. Dendrimer-Based Active Targeting of Colon Cancer Cells

The passive targeting strategy or the EPR effect is only relevant to static cancers that are very permeable. However, colon cancer cells in advanced stages have relatively poor or uneven permeability over their whole heterogeneous tumor population. Only the active targeting strategy, which permits the conjugation of several cancer-targeting ligands, may address these problems. To enable their binding specifically to overexpressed receptors on certain types of cancer cells, it is required to conjugate particular targeting ligands on nanocarrier surfaces and terminal functional groups in the case of dendrimers [96]. Various dendrimers functionalized with ligands for the selective delivery of anticancer drugs to CC cells are presented in Table 1.

Targeted dendrimers can be designed with different classes of targeting moieties, including small molecules, antibodies, peptides, and aptamers. These ligands are usually complementary in terms of binding to certain biomarkers that are particularly expressed in the case of CC. Small molecules include sugars (galactose, fructose, fucose, inulin) and vitamins (folate, biotin). There are certain tumor-associated antigens (TAAs) that are overexpressed in cancer cells and could be targeted via monoclonal antibody (mAb). For instance, in CC, EGFR serves as TAA and is usually targeted by mAbs, i.e., panitumumab and cetuximab, which have shown a success rate of 36 percent. Similarly, bevacizumab is another anti-vascular endothelial growth factor (VEGF) mAb used for CC targeting. Anti-programmed cell death protein 1 (PD1) mAbs pembrolizumab and nivolumab are specific to mismatch repair (MMR)-deficient CC and efficient in 30–50 percent of patients [97]. However, mAb-dependent targeting has certain limitations. Due to their heavy molecular weight and large size, they slowly diffuse and deliver into tumor tissues. The poor durability of mAb, whose functions depend on their intact spatial structure, is another limitation. Thus, in comparison to mAb, peptides are examined to be better for active targeting of dendrimer-based nanocarriers. Several peptides bind to tumor-specific targets, and these peptides are often categorized into three groups: first which attach to surface receptors of the cell, second which attach to the internal receptor of the cell and third are one which attach to the receptor present in the extracellular space. Aptamers are other well-established targeting molecules having specificity and affinity to a particular receptor only, similar to monoclonal antibodies. Aptamers have little immunogenic effects and are tiny enough to infiltrate deeply inside tumors. Aptamers have many advantages over antibodies, having excellent stability, simple synthesis, reduced batch-to-batch variation, and ease of chemical modifications that enable various types of conjugation.

PPL has low solubility in water; therefore, dendrimers have been used to increase its solubility in a trial. However, it has been noted that plain dendrimers, particularly higher-generation dendrimers, can be hazardous at certain doses. Hence, PEG-functionalized G4 PAMAM dendrimers were created using a click chemistry technique. The PEGylated G4 PAMAM dendrimers increased PPL’s solubility and enabled the persistent release of encapsulated PPL. Further, dendrimers-based formulation elevated apoptosis in human colon cancer cells more than pure PPL [98].

Alibolandi et al. developed targeted gold-dendrimer hybrid NPs (PEG-AuPAMAM) to deliver CUR. They examined the theranostic potential of a hybrid CUR-loaded gold-dendrimer (PEG-AuPAMAM-CUR) system. PAMAM dendrimers being highly branched macromolecules were used to stabilize gold NPs. CUR typically has restricted bioavailability due to its low water solubility (0.6 g/mL), poor absorption, high first-pass metabolism, rapid excretion, and degradation in alkaline medium [99]. By combining G5 PAMAM dendrimer with PEGylated amine-terminated tetrarchloroaurate ions, a dendrimer-gold hybrid structure was synthesized and loaded with CUR. The final hybrid system had a high payload of CUR. To target the mucin (MUC) 1-expressing CC, the CUR-loaded gold-coated-PEGylated dendrimer was functionalized with a MUC-1 aptamer. The results revealed that elevated cellular uptake and the accumulation of MUC-1 functionalized dendrimers (Apt-PEG-AuPAMAM-CUR) in HT29 and C26 cells caused higher cytotoxicity than in the non-targeted system. In addition, the accumulation of gold-hybridized dendrimer provides precise CT images of mice induced with C26 tumor- [100].

Similarly, Alibolandi et al. created a pegylated PAMAM dendrimer and loaded it with camptothecin (CPA. For site-specific targeting to CC cells overexpressing nucleolin receptors, they functionalized the dendrimers with AS1411 anti-nucleolin aptamers. An MTT assay showed that the nucleolin-positive HT29 and C26 colorectal cancer cell lines were more sensitive to the targeted CPA-loaded pegylated dendrimers than the nucleolin-negative CHO cell line. The identical technique was successfully tested in vivo on mice with C26 tumors [101].

IR is a common anticancer medication for CC treatment. However, due to its low therapeutic index, its usefulness is constrained. England et al. demonstrated improvement in the anticancer potency of SN-38, the biological intermediate of IR, using dendrimers. They modified the G5 PLL dendrimer with polyoxazoline. The conjugate produced persistent SN-38 level in systemic circulation higher than the desired dose with a circulation time of 21 h [102].

Narmani et al. tried to improve the specificity of OX using G4 PAMAM surface-grafted with folic acid as a small targeting molecule and PEG. PEG was conjugated to boost stability and half-life. In the SW480 cell line, it was demonstrated that the OX-loaded PEG-PAMAM complex had a greater cellular absorption. The effects of PEG-PAMAM-FA-OX on the suppression and proliferation of CC cells were seen in the cell viability assays [103].

**Table 1 pharmaceutics-15-00801-t001:** Recent trials in dendrimer-mediated targeting of anticancer drugs for the treatment of colon cancer.

Dendrimer	Drug	Delivery Mode	Targeting Ligand	Remarks	Ref.
G4 PAMAM	Capcitabine	Passive	-	Improve therapeutic index. Reduced side effects on liver and bone marrow.	[95]
G4 PAMAM	Piperlongumine	Passive	PEG	Improved aqueous solubility and persistent drug release. Reduced nonspecific hemolysis. Elevated colon cancer cell apoptosis.	[98]
Pegylated gold-coated G5 PAMAM	Curcumin	Active	MUC-1 aptamer	Higher cellular uptake and cytotoxicity in colon cancer cells. More precise CT imaging.	[100]
PAMAM	Camptothecin	Active	Anti-nucleolin AS1411-aptamer	Selective cytotoxicity in nucleolin-positive cell lines compared to nucleolin-negative cell lines.	[101]
G5 PLL	Irinotecan	Active	Polyoxazoline	Controlled drug release and prolonged circulation time. Reduced nonspecific gastrointestinal toxicity.	[102]
G4 PAMAM	Oxaliplatin	Active	Folic acid	Increased half-life and reduced immunogenicity. Improved cellular uptake and cytotoxicity in colon cancer cells.	[103]

## 6. Conclusions and Future Perspective

Despite having intrinsic anticancer potential, the majority of CC drugs do not perform sufficiently enough to completely sweep off the tumor cells and cause toxicity to healthy cells. These drawbacks of conventional drug therapy are related to the low solubility, non-specificity, less stability, short circulation span, nonspecific biodistribution, and rapid elimination of drugs in their native form.

Dendrimers are rapidly becoming a desirable type of drug delivery vector for cancer treatment because of recent breakthroughs in nanotechnology. Dendrimers are multipurpose smart nanocarriers that may safely and specifically deliver one or more therapeutic agents to cancer cells. Dendrimers are the most suitable nanocarriers for anticancer drug delivery due to their unique physicochemical properties. Dendrimers are used therapeutically for drug encapsulation, drug solubilization, cytotoxicity, blood plasma retention duration, biodistribution, and tumor uptake. Functional groups present in the exterior surface of dendrimers also make it possible to incorporate additional moieties that actively target CC, which is now often employed as a CC targeting technique. Passive targeting plays an equal role in achieving CC-specific therapy using dendrimers. The efficacy of the anticancer drug seems to increase significantly due to the selectivity improvement attributed to dendrimers, while the occurrence of negative effects on the normal cell decreases. Although dendrimer-based anticancer formulations have shown several benefits over the plain drug in vitro and preclinical studies, clinical developments are limited, probably requiring more preclinical studies.

## Figures and Tables

**Figure 1 pharmaceutics-15-00801-f001:**
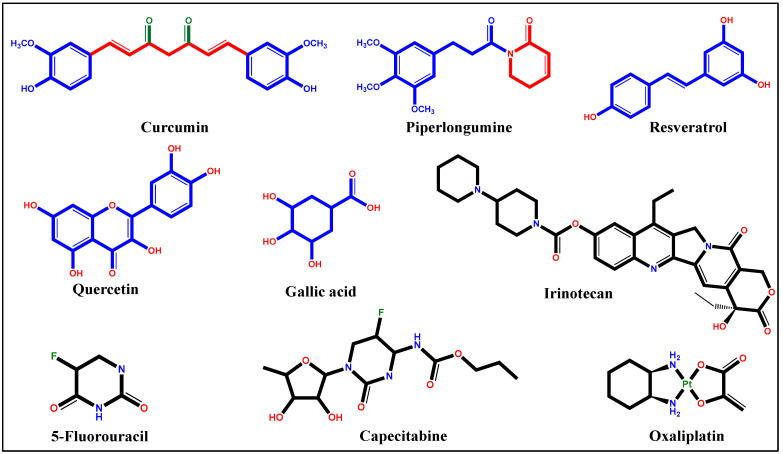
Structure of anticancer drugs used in the treatment of colon cancer.

**Figure 2 pharmaceutics-15-00801-f002:**
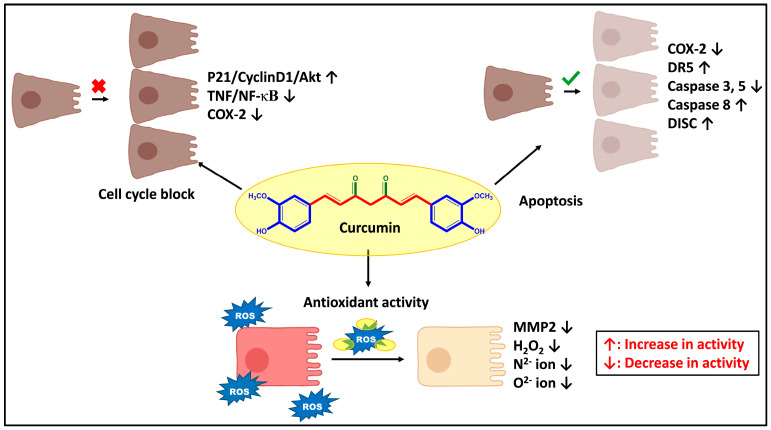
Anticancer mechanism of Curcumin in colon cancer.

**Figure 3 pharmaceutics-15-00801-f003:**
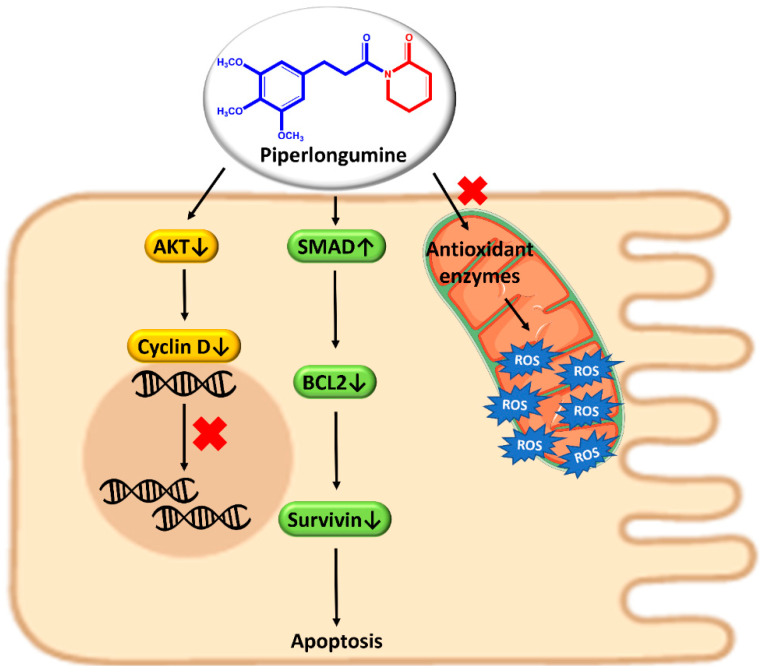
Anticancer mechanism of piperlongumine in colon cancer.

**Figure 4 pharmaceutics-15-00801-f004:**
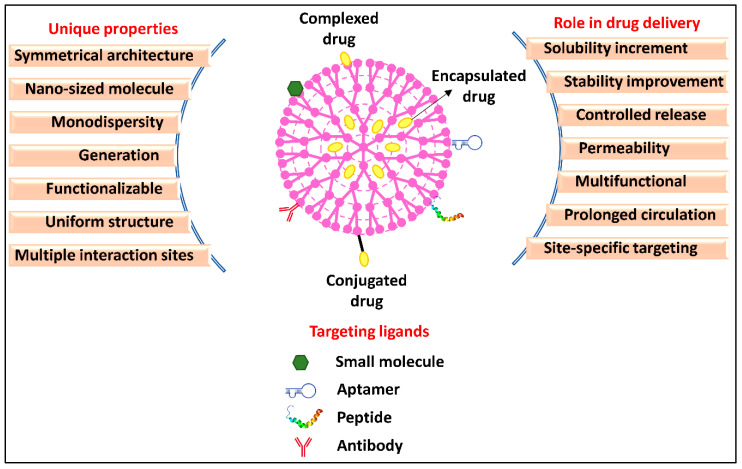
Dendrimer as a nanocarrier for anticancer drug delivery.

**Figure 5 pharmaceutics-15-00801-f005:**
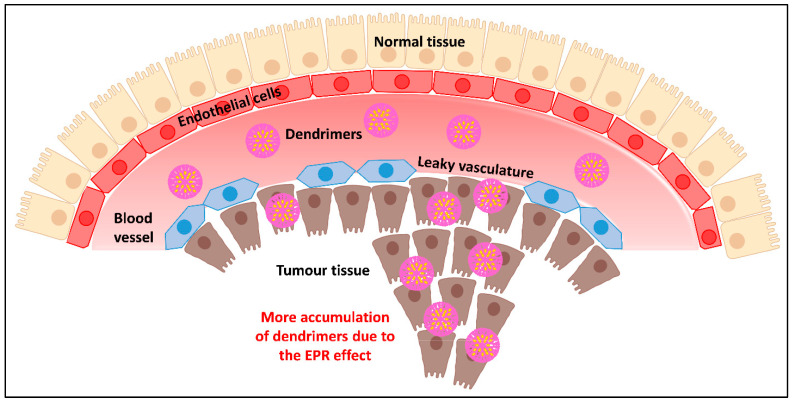
Dendrimer-mediated passive targeting of anticancer drugs.

## Data Availability

No new research data were created in this article, data sharing is not applicable for this article.

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
