# Peer review of "Dendrimer-Mediated Delivery of Anticancer Drugs for Colon Cancer Treatment"

_pharmaceutics, 2023, doi:10.3390/pharmaceutics15030801_

Round 1

Reviewer 1 Report

This is a nice review about dendrimer-mediated delivery of anticancer drugs for colon cancer treatment. I recommend it for publication after the following points are addressed.

1. 'Nanoparticles (NPs) have been clinically used as smart nanocarriers for localized CC chemotherapy, additionally lowering the doses, increasing the drug's aqueous solubility and bioavailability with fewer side effects', several studies (Biomacromolecules 17 (6), 2010-2018; Pharmaceutics 2022, 14(11), 2400; Acta Biomaterialia 10.12 (2014): 5116-5127) are recommended to be included to support such a claim.

2. Copyright permission statements should be added.

3. Most of the time, high drug loading capacity can't be achieved in the dendrimer system.

4. It is not convincing that the prolonged circulation is only due to the tiny size of the dendrimer.

5. Techical issues. '[12.]' to '[12].' Please check all.

Author Response

This is a nice review about dendrimer-mediated delivery of anticancer drugs for colon cancer treatment. I recommend it for publication after the following points are addressed.

  1. 'Nanoparticles (NPs) have been clinically used as smart nanocarriers for localized CC chemotherapy, additionally lowering the doses, increasing the drug's aqueous solubility and bioavailability with fewer side effects', several studies (Biomacromolecules 17 (6), 2010-2018; Pharmaceutics 2022, 14(11), 2400; Acta Biomaterialia 10.12 (2014): 5116-5127) are recommended to be included to support such a claim.

Response 1: The suggested articles have been incorporated as references no. 4 to 6, in the support of the aforementioned portion of the introduction section.

  1. Copyright permission statements should be added.

Response 2: All the figures incorporated in the manuscript have been originally drawn.

  1. Most of the time, high drug loading capacity can't be achieved in the dendrimer system.

Response 3: We are thankful to the reviewer for his keen observation. The 3.3. point mentioning the high drug loading capacity of the dendrimer system has been removed from section 3.

  1. It is not convincing that the prolonged circulation is only due to the tiny size of the dendrimer.

Response 4: Other factors and strategies to achieve prolonged circulation using the dendrimer system have been added to point (v).

  1. Technical issues. '[12.]' to '[12].' Please check all.

Response 5: The technical issue in the in-text citation [12] has been corrected and all other in-text references have been checked.

Reviewer 2 Report

The manuscript "Dendrimer-mediated delivery of anticancer drugs for colon cancer treatment" is good piece of work, but different in quality between point 2: "Anticancer drugs.... and points 3, 4 and 5" Role of dendrimers..... and "Dendrimer-based......

In my opinion point 2 describing the properties of antitumor agents against colon cancer is very well written with well prepared figures.

It is in contrast to points related to dendrimers. There are long piece of the text without any figures and schemes. It is difficult to accept this form, because there were described a lot of dendrimer structures, which are "....in a huge variety" and their names as PAMAM, PPI, PLL and others are not friendly for the reader, who would like to found in this work a wide knowledge in the field of dendrimers. Part 3 contains also the list of advantages in anticancer drug delivery. This seems to be a good idea, however, it should be numbered 1, 2, 3.... not 3.1, 3.2, 3.3...Moreover, I propose the short graph for each point.

Taking into account points 4 and 5, the figure 6 is not clear representation of the texts. There should be illustrated different anatomy between normal and tumor tissue in relation to drug loaded in different types of dendrimers.

Author Response

  1. The manuscript "Dendrimer-mediated delivery of anticancer drugs for colon cancer treatment" is good piece of work, but different in quality between point 2: "Anticancer drugs.... and points 3, 4 and 5" Role of dendrimers..... and "Dendrimer-based......In my opinion point 2 describing the properties of antitumor agents against colon cancer is very well written with well-prepared figures. It is in contrast to points related to dendrimers. There are long piece of the text without any figures and schemes. It is difficult to accept this form, because there were described a lot of dendrimer structures, which are "....in a huge variety" and their names as PAMAM, PPI, PLL and others are not friendly for the reader, who would like to found in this work a wide knowledge in the field of dendrimers.

Response 1: Figure 4 demonstrating the unique properties of dendrimer as a nanocarrier and its role in drug delivery has been incorporated at the end of the section 3 of the manuscript.

  1. Part 3 contains also the list of advantages in anticancer drug delivery. This seems to be a good idea, however, it should be numbered 1, 2, 3.... not 3.1, 3.2, 3.3...Moreover, I propose the short graph for each point.

Response 2: The points in section 3 listing the advantages of dendrimers in anticancer drug delivery have been numbered i, ii, iii……instead of .1, 3.2, 3.3...Also, each point is expanded by illustrating the example for each point.

  1. Taking into account points 4 and 5, the figure 6 is not clear representation of the texts. There should be illustrated different anatomy between normal and tumor tissue in relation to drug loaded in different types of dendrimers.

Response 3: Figure 5 has been incorporated to illustrate the different anatomy between normal and tumor tissue in relation to the passive drug delivery mediated by the dendrimers.
